# Analysis of Variation in Distance, Number, and Distribution of Spotting in Southeast Australian Wildfires

**Michael A. Storey** [1],*[ID], **Owen F. Price** [1], **Ross A. Bradstock** [1] **and Jason J. Sharples** [2,3]

1   Centre for Environmental Risk Management of Bushfires, University of Wollongong, Wollongong, NSW 2522, Australia; oprice@uow.edu.au (O.F.P.); rossb@uow.edu.au (R.A.B.)
2   School of Science, University of New South Wales (UNSW), Canberra, ACT 2600, Australia; j.sharples@adfa.edu.au
3   Bushfire and Natural Hazards Cooperative Research Centre, Melbourne, VIC 3002, Australia
*   Correspondence: mas828@uowmail.edu.au

**Abstract:** Spotting during wildfires can significantly influence the way wildfires spread and reduce the chances of successful containment by fire crews. However, there is little published empirical evidence of the phenomenon. In this study, we have analysed spotting patterns observed from 251 wildfires from a database of over 8000 aerial line scan images capturing active wildfire across mainland southeast Australia between 2002 and 2018. The images were used to measure spot fire numbers, number of "long-distance" spot fires (> 500 m), and maximum spotting distance. We describe three types of spotting distance distributions, compare patterns among different regions of southeast Australia, and associate these with broad measures of rainfall, elevation, and fuel type. We found a relatively high correlation between spotting distance and numbers; however, there were also several cases of wildfires with low spot fire numbers producing very long-distance spot fires. Most long-distance spotting was associated with a "multi-modal" distribution type, where high numbers of spot fires ignite close to the source fire and isolated or small clumps of spot fires ignite at longer distances. The multi-modal distribution suggests that current models of spotting distance, which typically follow an exponential-shaped distribution, could underestimate long-distance spotting. We also found considerable regional variation in spotting phenomena that may be associated with significant variation in rainfall, topographic ruggedness, and fuel descriptors. East Victoria was the most spot-fire-prone of the regions, particularly in terms of long-distance spotting.

**Keywords:** spot fire; spotting distance; spotting distribution; wildfire behaviour

---

## 1. Introduction

There are several behaviours known to increase the destructive and chaotic nature of wildfires [1]. One of the most important is spotting, a stochastic form of wildfire spread that leads to the ignition of spot fires: Separate new fires ignited by firebrands (burning pieces of vegetation or other flammable material) originating from a wildfire. Under intense spotting, wildfire spread becomes highly unpredictable and impossible for fire crews to contain [2].

Spotting is controlled by several variables that are known to influence one or multiple stages of the spotting process (i.e., firebrand generation, firebrand transport, and the ignition of receiver fuels [2]) and the subsequent role of spotting in overall wildfire spread. This includes fuel (amount, bark type), weather (moisture, wind), topography (slope steepness, ridges), and main fire (or "source fire") attributes (size, intensity, plume characteristics) [2–9]. As these factors vary both locally (e.g., ridge vs. valley) and regionally (e.g., damp mountain forests vs. dry grassy plains,

eucalypt forests vs. conifer forests), it can be expected that local and regional variation in spotting patterns also exist.

More extreme cases of spotting are characterised by long distances (up to many kilometres) and/or the ignition of tens or hundreds of spot fires in mass spotting events [10,11]. Interaction between multiple spot fires and the source fire potentially increases burning intensity, flame heights [12], and wildfire spread rates over large areas [11,13]. Such interactions are likely to depend on the distribution of spot fires (number and distances); for example, multiple spot fires in close proximity (to the source fire and/or other spot fires) are likely to interact, whereas isolated spot fires are not. While extreme spotting occurs in some wildfires, other wildfires only produce mild spotting behaviour (e.g., low number, short distance) that has little impact on overall spread; for example, one or two short-distance spot fires may be quickly overrun by the source fire [14]. While significant understanding has been gained in aspects of spotting [2], studies that describe observed spotting patterns, including variation in spotting distances, numbers, spotting distributions, and geographic variation in spotting patterns from a large sample of real wildfires, are lacking.

Current understanding mainly stems from laboratory or mathematical modelling of firebrands, anecdotal sources (e.g., firefighter experiences), and agency reports or published studies on behaviour of individual or a few wildfires, often with only general descriptions of spotting (e.g., "dense" spotting) [10,15,16]. A significant recent study looked at a large sample of real wildfires from a single season in the Northern Rockies, which included 48 wildfires tracked over multiple days and 7214 spot fires identified [6]. A total of 94% of spot fires were within 500 m, with rarer long-distance spotting to a maximum distance of 2.7 km, with the most significant driver being an interaction between wind speed and fire growth (from previous day). Storey et al. [3] modelled drivers of spotting distance and numbers, finding source fire area to be the best predictor, but did not report on spotting distributions. In southwest Western Australia, the Project Vesta experiments measured spot fire distributions (i.e., the downwind distribution of spot fires at different distances from the source fire at any one time) from a series of controlled burns [17,18], four of which had firebrand distribution data recorded and five had spot fire distances recorded. The fires produced approximately exponential distributions of firebrands and spot fires, with most igniting close to their source fires (< 50 m). The total number of spot fires was correlated with plume behaviour, fuel characteristics (hazard score, age), and wildfire behaviour (rate of spread, intensity, flame height). There have been several international studies focused on determining firebrand landing distributions based on small-scale laboratory experiments and/or mathematical modelling [19–27]. These have generally indicated right-skewed distributions of firebrands (e.g., exponential, Rayleigh, lognormal), but normal or bi-modal distributions have also been reported. The distributions found are influenced by factors such as firebrand size, shape, burning rate, lofting height, and wind speed. The applicability of these models to spotting distributions (i.e., including spot fire ignition [9]) from real wildfires may be limited as, for example, models usually do not consider large highly aerodynamic firebrands (e.g., eucalypt ribbon bark [4]), wildfires burning on rugged terrain, or the effects of fire–atmosphere interaction. To validate and improve such models, comparison to spotting distributions from real wildfires is needed.

Southeast Australia is generally acknowledged as being prone to large wildfires with intense spotting behaviour (Figure 1) [16]. Spotting has played a significant part in some of the most dangerous and destructive wildfires in southeast Australia, including the 2009 Black Saturday Fires, where one wildfire produced spot fires between 30 and 35 km away [10,28]. Other examples include the 2003 Alpine and Canberra wildfires [29], multiple wildfires in the Blue Mountains near Sydney (including 2013, 2006, 2001/2), the 2013 Wambelong wildfire [30], and the 1983 Ash Wednesday wildfires [15]. Southeast Australia experiences dry summers (including periodic droughts) and days where high temperatures, low humidity, and strong winds can lead to the development of extreme wildfires and associated spotting [31]. Wildfires in southeast Australia with high degrees of spotting are generally associated with large forested areas dominated by Eucalypt species with fibrous or flaky

bark types that produce ideal firebrand fuel [4,5,32]. However, the significant variation in climate and vegetation that exists, from tall moist forests in rugged terrain to dry and sparsely vegetated desert plains, suggests that significant variation in spotting also exists. Published reports of spotting patterns have historically been limited, mainly due to limitations in technology to record spotting. However, since the early 2000s, fire agencies in southeast Australia have regularly been deploying aircraft fitted with multispectral/infrared line scanners [3,33,34] to capture detailed images of hundreds of wildfires. This has created a large archive of real-world observations of wildfires and spotting. In this study, we use this line scan imagery to measure and describe spotting patterns across mainland southeast Australia. We compare simple measures of wildfire spotting (maximum distance, number of spot fires) across five broadly defined regions of southeast Australia for the period 2002 to 2018, and also describe different types of spotting distance distributions. In particular, we use the data to explore the following sets of questions about spot fire numbers, distances, and regional variation:

1. How many spot fires are commonly produced in wildfires? How common is long-distance and mass spotting? What is the correlation between long-distance spotting and mass spotting (high numbers)?
2. How are spot fires commonly distributed (mostly near or far)? Is there significant variation in spotting distributions?
3. Is there significant geographic variation in spotting? Is it more common in some regions than others?

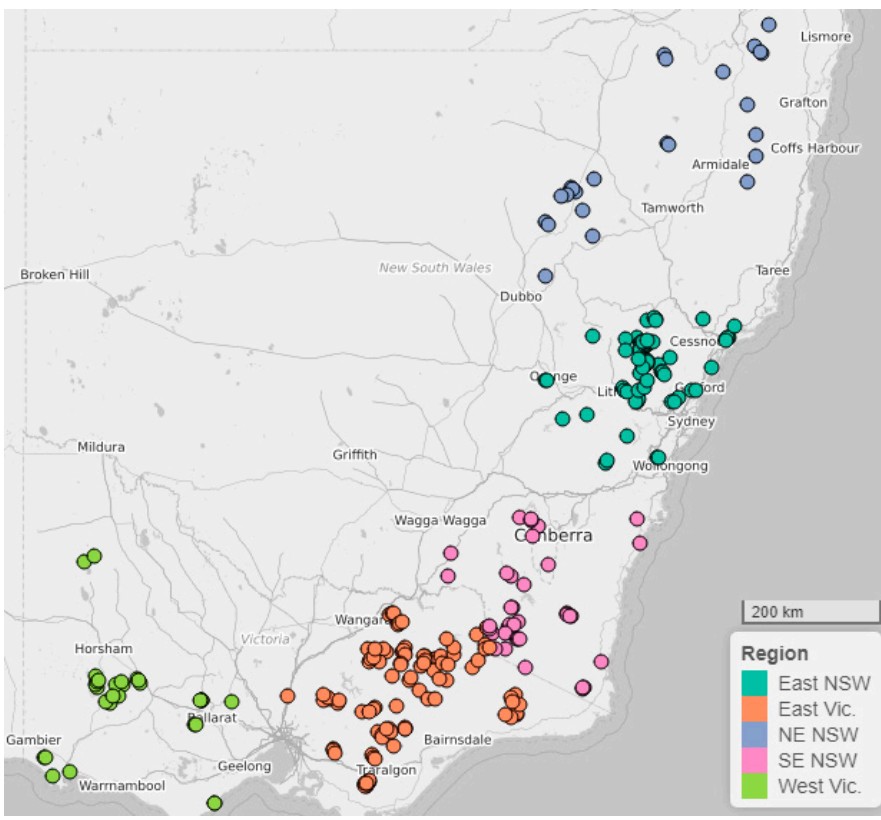

**Figure 1.** Locations of source fires captured and analysed for this study, coloured by assigned region. Base-map © OpenStreetMap contributors.

## 2. Materials and Methods

### 2.1. Study Location and Line Scans

We focused on wildfires on mainland southeast Australia between 4 Dec 2002 and 28 Feb 2018 (Figure 1). Since ~2001, the New South Wales Rural Fire Service (RFS, operated by Air Affairs Australia Pty Ltd) and Victorian Department of Environment, Land, Water, and Planning (DELWP) have regularly deployed aircraft fitted with optical line scanning equipment to wildfires to capture georectified images for use in wildfire response operations. When present, actively burning wildfire areas including spot fires can be clearly identified in the images. Over 15,000 individual images that show whole or part of a wildfire, from actively burning to extinguished, were provided by RFS and DELWP.

The scanner captures radiance emitted from wildfires in multiple spectral bands, including smoke-penetrating infrared, by continuously scanning side-to-side as the aircraft moves forward, from which a georectified image is created [34]. RFS uses a subset of three bands that are extracted to create images that display active fire as yellow-orange (Figures 2 and 3). Currently, two shortwave infrared (SWIR; at 1.6 and 2.2 µm) and one blue band are usually used, although different band combinations, such as thermal infrared, SWIR, and green, have been used in the past/depending on fire conditions [35]. DELWP creates single-band images (longwave infrared at 8–14 µm) that are processed to highlight active fire (high longwave infrared pixel values) as red (Figure 4).

We found that ~10% of all images supplied had active source fires with at least one spot fire (~50% of all images had active fires but no spot fires). Images that contained active fires with zero spot fires, no active fire, flank-only fires, or image acquisition errors (e.g., colour saturation) were not used in this analysis. The line scan operator can adjust sensitivity to different spectral bands on each mission, but the sensitivity settings were not recorded in the data. This may have affected spot fire detectability in some images.

In cases where two scans showed the same spot fires (i.e., scans of the same source fire acquired within a short period), the image that showed the longest distance downwind from the source fire was retained and the other image was excluded. This was to ensure spot fires were not measured twice. Several images contained georectification errors, and some images were not georectified. In these cases, georeferencing of the images to a common base image was carried out in ArcGIS 10.5.

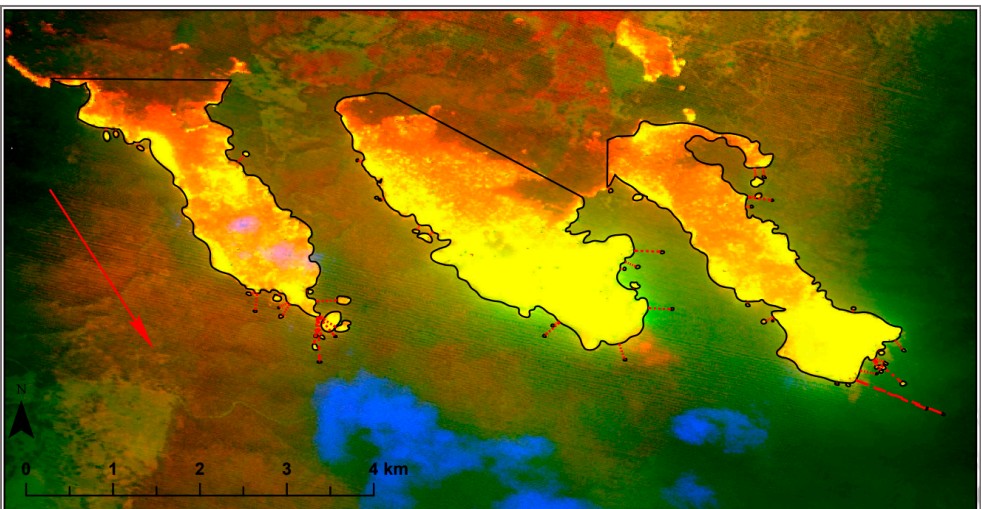

**Figure 2.** Rural Fire Service (RFS) line scan showing three separate source fires (three largest polygons). Most actively burning fire is yellow, orange is still hot after the main fire front has passed, brown is extinguished, green is unburnt vegetation, and blue is faint signature of the smoke plume. Red dotted lines indicate spot fire (small polygons) distances measured for analysis. The red arrow indicates spread direction.

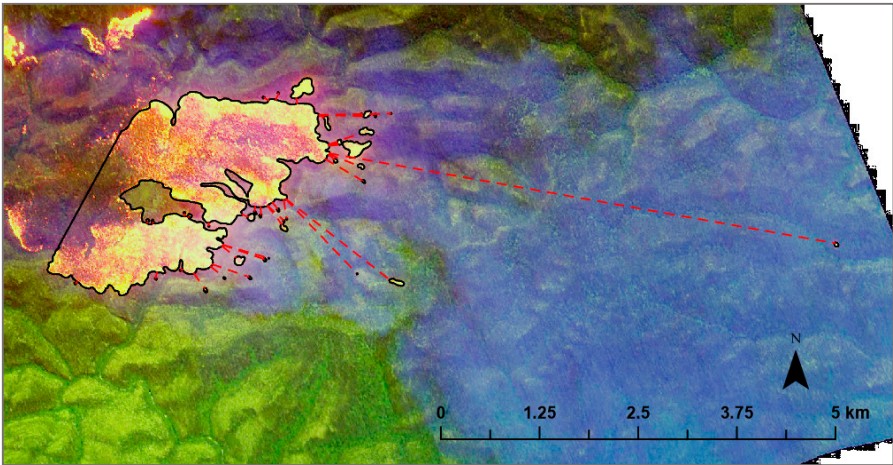

**Figure 3.** RFS line scan showing individual source fire (largest polygon) and spot fires (small polygons). Interpretation as in Figure 2.

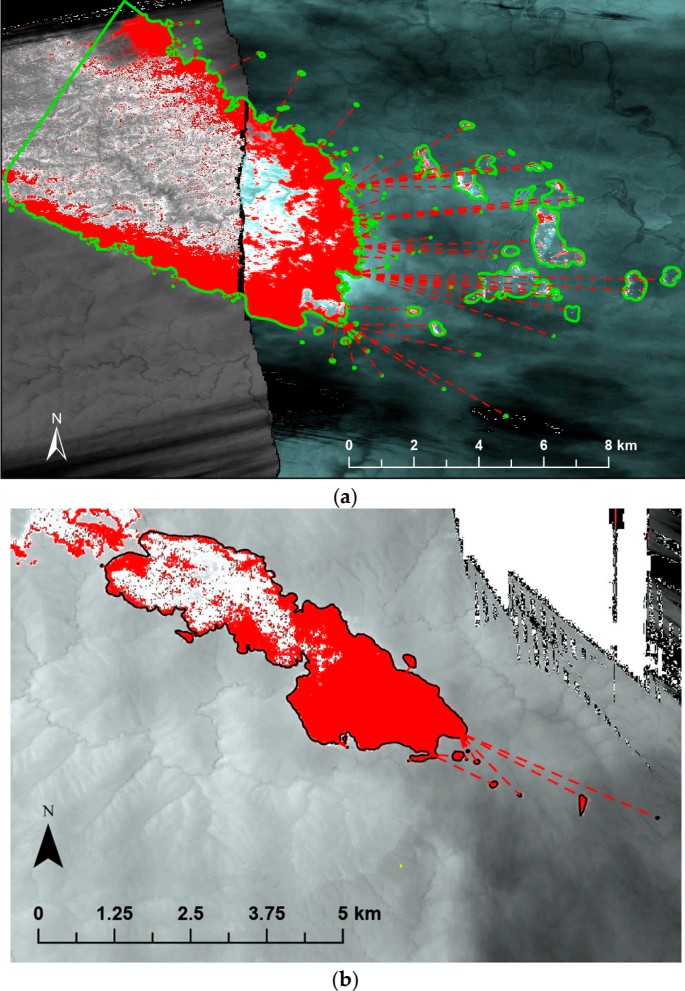

**Figure 4.** Department of Environment, Land, Water, and Planning (DELWP) line scan examples. Actively burning fire is processed to show as red. White is recently burnt, light to dark greys are unburnt vegetation. Black areas are dense parts of the smoke plume or cloud. Red dashed lines indicate spot-fire distance measured for analysis. The largest polygons represent source fire, the smallest polygons are spot fires (green in (**a**), black in (**b**). (**b**) has some larger spot fires in grassland. See other examples in Storey et al. [3].

## 2.2. Mapping Fires

GIS (Geographic Information System) polygons were created for sampling in ArcGIS 10.5 by manually digitizing all spot fires and their source fire in each image. The source fire (meaning source of firebrands) was defined as the actively burning part of the main wildfire, which was a head-fire burning separately from any other head-fires in the image (e.g., two tongues of fire spreading along separate ridges were different source fires, Figure 2). A single source fire could be digitized from a mosaic of several images captured sequentially (each image captured only part of the source fire; Figure 4), or multiple source fires could be digitized from a single image (Figure 2). The source fire polygon was digitized by identifying an actively burning head-fire and its spread direction (based on its shape), then drawing a polygon from the head-fire tip and back along the active fire length of each flank before finalizing the polygon with a line joining the back end of the two flanks (Figure 3). Spot fires were each digitized as separate polygons and assigned to the closest source fire.

Spot fire distance to the closest point on the edge of the source fire was measured using the ArcGIS 10.5 "Near" tool. We assumed that all spot fires originated from the source fire polygon, although there is a chance that that some spot fires were ignited by other nearby spot fires. We also assumed all spot fires in the vicinity were captured in each image, although it is possible some spot fires were missed because they were very small, beyond the downwind image extent, or were under dense smoke or thick canopy (also see Storey et al. [3] for further details).

## 2.3. Sampling and Analysis

There were 338 source fires with at least one spot fire identified from the line scan images. However, in order to analyse spotting distribution patterns, we used only those source fires with at least three spot fires (n = 251). The 251 source fires included spanned a wide range of fire weather and behaviour (for all fires, wind gusts (km/h) ranged from 14 to 81 (mean 37), temperature ranged from (Celsius) 12 to 42 (mean 28), relative humidity (%) ranged from 7 to 63 (mean 27) [3]).

Source fires were grouped into five regions based on latitude, longitude, and state borders (Figure 1): East Victoria (N = 94), southeast New South Wales (SE NSW, n = 37), east New South Wales (n = 66), west Victoria (n = 30), and northeast New South Wales (NE NSW, n = 24). Regional summary values of elevation [36], canopy height [37], and annual rainfall [38] were calculated by: (1) Sampling rasters of each variable with 50 m point grids covering each source fire polygon, (2) calculating source fire summary statistics, and (3) calculating mean regional values from source fire summary values. Mean values were calculated for all variables. Standard deviation of elevation within each source fire was also calculated as a measure of ruggedness. As we sampled from source fires, our summary values do not necessarily represent the entire region, but rather those locations where fires occurred within that region.

In addition, we identified predominant spotting-prone bark types across our source fire areas [4,5,39]. Identification of bark type relied on state-based vegetation maps that list common species in each forest class. We assigned bark types by (1) sampling state-based vegetation maps (Ecological Vegetation Classes in Victoria, Keith [40] vegetation classes in NSW) in each source fire area using a 50 m point grid, (2) based on the vegetation class description of dominant/most common species, assigning each vegetation class to a category of stringybark dominant, ribbon bark dominant, shared stringy and ribbon bark dominance, or other bark type dominance (e.g., box-ironbark forests, shrubland, *Callitris* spp., *Angophra* spp.), and (3) summarizing the proportion of each bark type by region. Some vegetation types classified as one bark type may contain minor components of other bark types. The identification of bark types could be improved if more detailed tree species mapping (or bark mapping) existed across the study area; however, we believe the method used here is sufficient to identify general trends.

Kruskal–Wallis tests [41] were run to test for differences among regions for maximum spotting distance, total number of spot fires (irrespective of distance), and number of spot fires > 500 m from the source fire (indicating more extreme spotting behaviour). Maximum spot fire (spotting) distance here

means the farthest spot fire from source fire in the line scan image, rather than maximum firebrand flight distance (e.g., in modelling [7,42–44]). Kruskal–Wallis tests were combined with post-hoc Wilcoxon tests [41] to identify significantly different region pairs. Spearman's ranked correlation coefficients ($r_s$) were also calculated between each combination of the three spotting measures (all regions pooled). This was to identify whether mass spotting and long-distance spotting usually occurred from the same source fires.

To investigate spot fire distance distributions, frequency distribution polygon plots (similar to histograms, but represented by lines) of spot fire distances (250 m bins) were plotted for each source fire. Based on visual inspection, each plot was assigned to one of three broad types of distribution based on its general appearance and the largest distance between any two spot fires. The distribution types were:

1. Exponential: Most spot fires within the first 250 m, decreasing numbers of spot fires in each bin out to the farthest spot fire and no gap > 1 km between any two spot fires (e.g., Appendix A, also the fires in Figure 2).
2. Multi-modal: Like type 1, except with a gap >= 1 km between any two spot fires (e.g., Appendix A, also the fire in Figure 3).
3. Other distribution: No prominent peak in the frequency plot, or the peak in spot fires was beyond the first 250 m bin (Appendix A).

Map examples of each distribution type are included in Appendix A.

The percentage of each distribution type in each region was calculated and Chi-Squared tests were used to indicate if spot fire distribution type was independent of region. Kruskal–Wallis and Wilcoxon tests were used to test for any significant differences between the distribution types in maximum spot fire distance, number of spot fires, or number of spot fires > 500 m (all regions pooled). See Supplementary Materials for data used in this analysis: Table S1) the 50 m sampling point grid, Table S2) a table of fuel, weather, and topography variables summarized by source fire and Table S3) a table containing all spot-fire distances.

## 3. Results

### 3.1. Spotting Distance Summary

Distances for a total of 4219 spot fires from the 251 source fires were analysed. The distribution of distance values (i.e., all spot fire distances from all source fires pooled) revealed that most spot fires were close to their source fire (mean = 0.4 km, median = 0.1 km; Figure 5). Long-distance spotting was much less common; only 10% of spot fires were beyond 1 km and 2% were beyond 3 km.

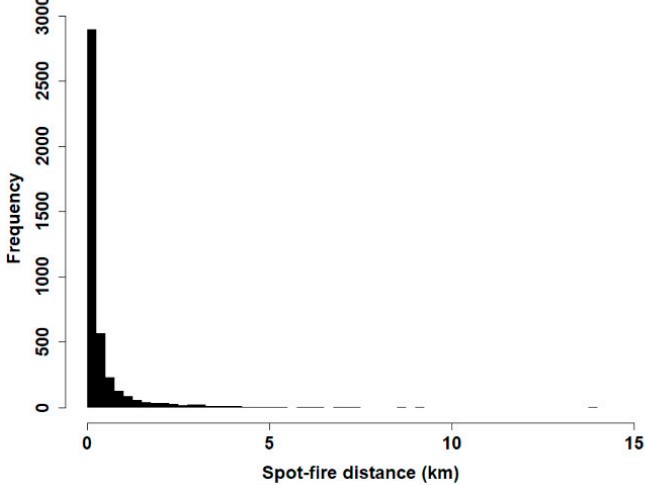

**Figure 5.** Distribution of distances of all spot fires (n = 4219) from every source fire pooled.

### 3.2. Correlation between Spotting Measures (All Data Pooled)

Maximum spot fire distance values ranged from 20 m to 13.9 km (mean 1.27 km, median = 0.64 km). Number of spot fires (N spot fires) ranged from three (the threshold for inclusion in the study) to 128 (mean = 16.81, median = 10) and number of spot fires > 500 m (N spot fires > 500 m) ranged from 0 to 55 (mean = 3.03, median = 1).

N spot fires > 500 m and maximum spot fire distance were strongly correlated ($r_s$ = 0.9). There was a weaker correlation between N spot fires > 500 m and N spot fires ($r_s$ = 0.67), and between N spot fires and maximum spot fire distance ($r_s$ = 0.62) (Figure 6). Despite a strong correlation, there were several source fires that spotted long distances but had few spot fires > 500 m. For example, there were seven source fires that produced at least one spot fire over 5 km but had <= 10 spot fires past 500 m. This included the longest spotting source fire (13.9 km, Wandoo 2006 fire in SE NSW), which had nine spot fires in total, and only five over 500 m.

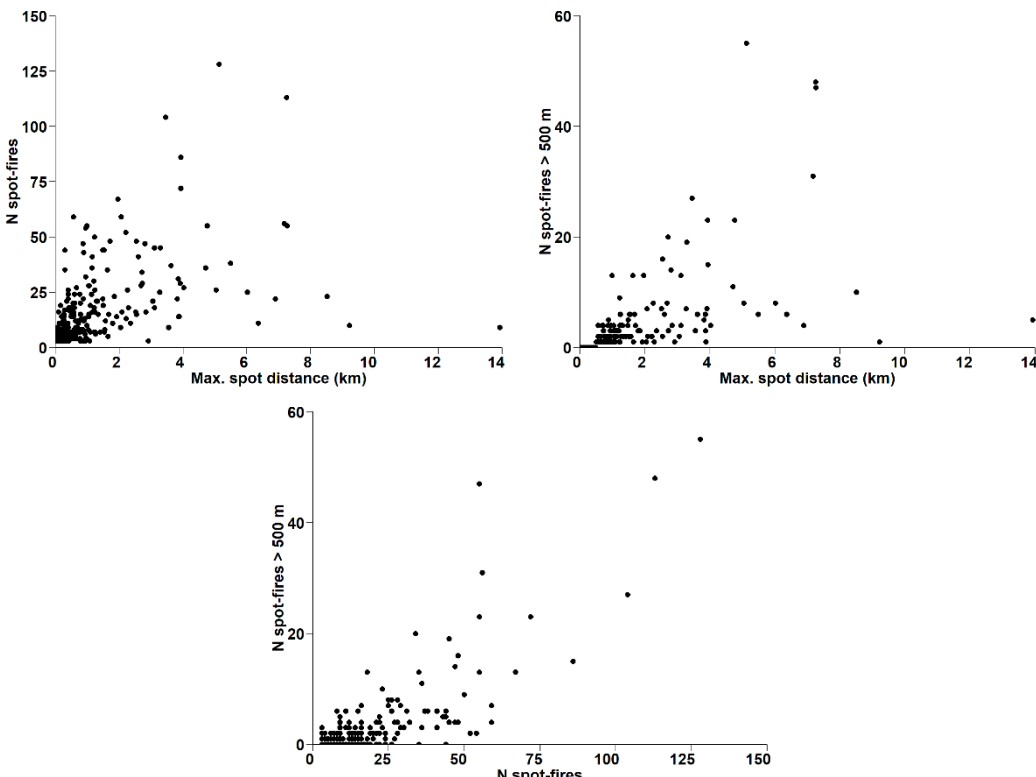

**Figure 6.** Scatterplots depicting pairwise relationships between the three spotting measures.

### 3.3. Spot Fire Distance Distribution Types (All Regions Pooled)

An exponential distribution was the most common distribution type across all regions (196 source fires), followed by multi-modal (39) and other (16). The spot fire distance distributions for each group are illustrated in Figure 7 (also see maps in Appendix A). For clarity, this plot was limited to the 10 source fires with the longest maximum distance in each group. Kruskal–Wallis and Wilcoxon tests indicated that the multi-modal group had significantly ($p$ < 0.01) higher maximum spot fire distances than the others (Table 1). Of the 11 source fires that spotted over 5 km, 10 were multi-modal and one was other-distribution. The exponential group had the lowest maximum distance values (Table 1) and contained high numbers of short-distance spot fires. The other-distribution group had low peaks in number of spot fires close to the source fire, but also had some longer-distance spot fires (Table 1, Figures 7 and 8). The height of the first initial peaks (i.e., number of spot fires in first 250 m from source fire) for the exponential and multi-modal groups ranged between 2 and 58, and between 2 and 11 for the other-distribution group.

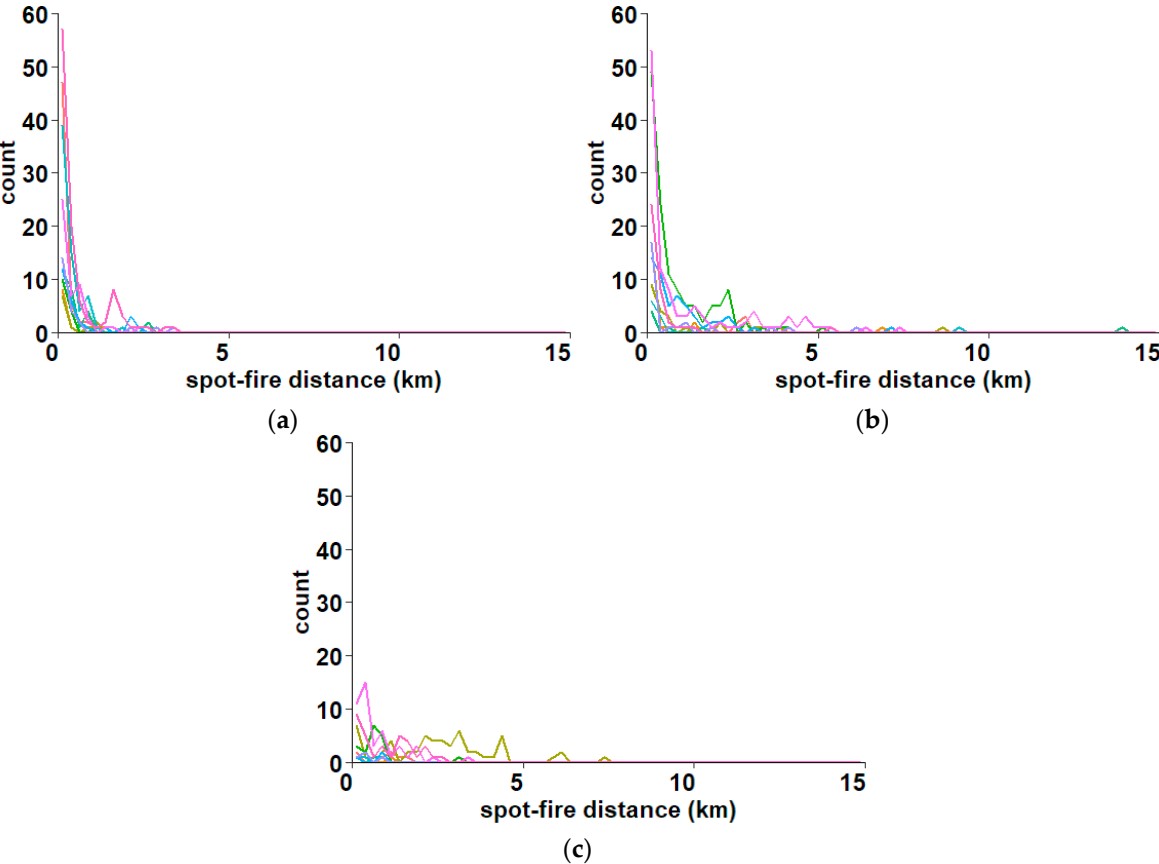

**Figure 7.** Frequency polygon distributions (bin width 250 m) for ten source fires with the longest maximum spot fire distance for each distribution type: Exponential group (**a**), multi-modal group (**b**), and other distributions (**c**).

**Table 1.** Summary statistics for the three spotting measures by distribution type.

|  | Multi-Modal | Exponential | Other |
|---|---|---|---|
| **N spots: Mean (median)** | 32.2 (23) | 14 (8) | 13.8 (7.5) |
| **N spots: Min–max** | 3–128 | 3–104 | 3–55 |
| **N spots > 500 m: Mean (median)** | 8.7 (4) | 1.5 (0) | 7.6 (2) |
| **N spots > 500 m: Min–max** | 1–55 | 0–27 | 0–47 |
| **Max distance km: Mean (median)** | 4.3 (3.8) | 0.6 (0.4) | 1.6 (0.9) |
| **Max distance km: Min–max** | 1.65–13.9 | 0.02–3.5 | 0.36–7.3 |

The multi-modal group had a significantly higher ($p < 0.01$) N spot fires than the exponential and other-distribution groups. The multi-modal group had a significantly higher ($p < 0.01$) N spot fires > 500 m than the exponential group, but not the other-distribution group.

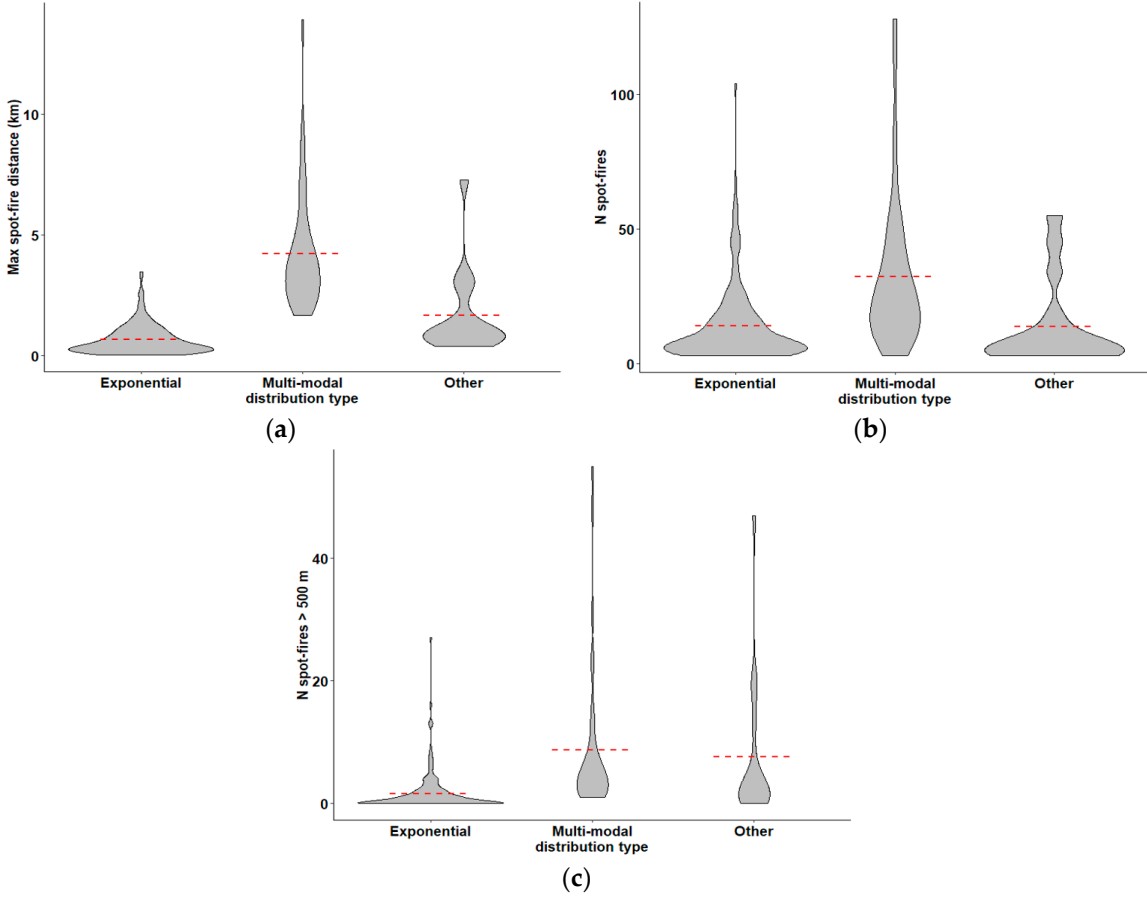

**Figure 8.** Violin plots of distribution type vs. maximum spot fire distance (**a**), number of spot fires (**b**), and number of spot fires > 500 m (**c**). The red dotted line is the mean group value.

### 3.4. Regional Spotting Distance and Number

Maps of the three spotting measures suggest areas of high spotting potential in east Victoria, SE NSW (close to the Victorian border), and around Sydney in east NSW (Figure 9). In particular, high N spot fires > 500 m and very long-distance spot fires mostly occurred in east Victoria or just over the state border in SE NSW. There were some source fires that produced very long-distance spot fires around Sydney in east NSW, but few source fires there had high N spot fires > 500 m. Long-distance spot fires and high values of N spot fires > 500 m were mostly absent from west Victoria and NE NSW in our data.

The patterns observed in the maps were supported by the statistical comparisons among regions. East Victoria and SE NSW had the highest group mean values for all spotting measures (Figure 10, Table 2). East NSW had the third largest mean for all spotting measures. East Victoria, east NSW, and SE NSW all had maximum values over 8 km, with the largest in SE NSW of 13.9 km. The maximum values for N spot fires (128) and N spot fires > 500 m (55) came from one source fire in east Victoria. NE NSW and west Victoria had the lowest mean values for all spotting measures.

Kruskal–Wallis tests indicated a significant difference between regions for all three of the spotting measures. Post-hoc Wilcoxon tests showed the most significant difference (lowest $p$ value) between east Victoria and NE NSW for maximum distance and N spot fires > 500m ($p < 0.01$) and between SE NSW and NE NSW for N spots ($p < 0.01$). There was no significant difference between SE NSW and east Victoria, or between NE NSW and west Victoria for any of the three spotting measures. East NSW had significantly higher ($p < 0.01$) spotting than NE NSW (excluding N spot fires ($p = 0.27$)) and significantly lower ($p < 0.05$) spotting than east Victoria for the three spotting measures.

**Table 2.** Summary statistics for the three spotting measures by region.

|  | East NSW | East Vic. | NE NSW | SE NSW | West Vic. |
|---|---|---|---|---|---|
| N spots: Mean (median) | 14 (8) | 20.1 (13.5) | 8.3 (6.5) | 22 (17) | 13.1 (8) |
| N spots: Min–max | 3–86 | 3–128 | 3–26 | 3–104 | 3–47 |
| N spots > 500 m: Mean (median) | 1.7 (1) | 5.1 (1) | 0.5 (0) | 3.6 (2) | 0.9 (0) |
| N spots > 500 m: Min–max | 0–15 | 0–55 | 0–8 | 0–27 | 0–4 |
| Max distance km: Mean (median) | 1 (0.6) | 1.8 (0.9) | 0.4 (0.3) | 1.6 (0.9) | 0.7 (0.3) |
| Max distance: Min–max | 0.08–9.2 | 0.06–8.5 | 0.02–2.3 | 0.04–13.9 | 0.02–4 |

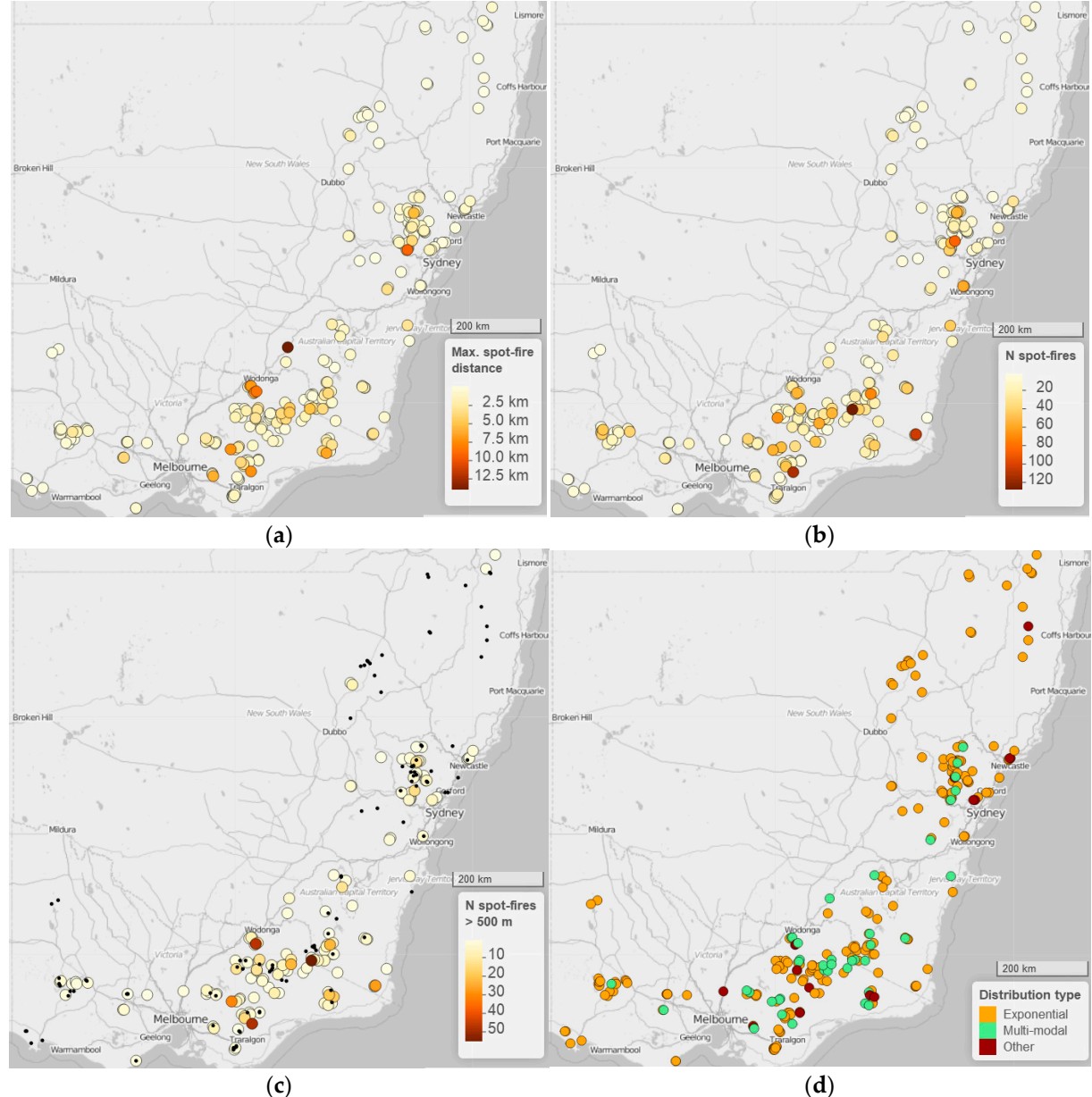

**Figure 9.** Points representing source fire locations coloured by maximum spot fire distance (**a**), number of spot fires (**b**), number of spot fires > 500 m (**c**), and distance distribution type (**d**) across study area. Black dots are zero values. Base-map © OpenStreetMap contributors.

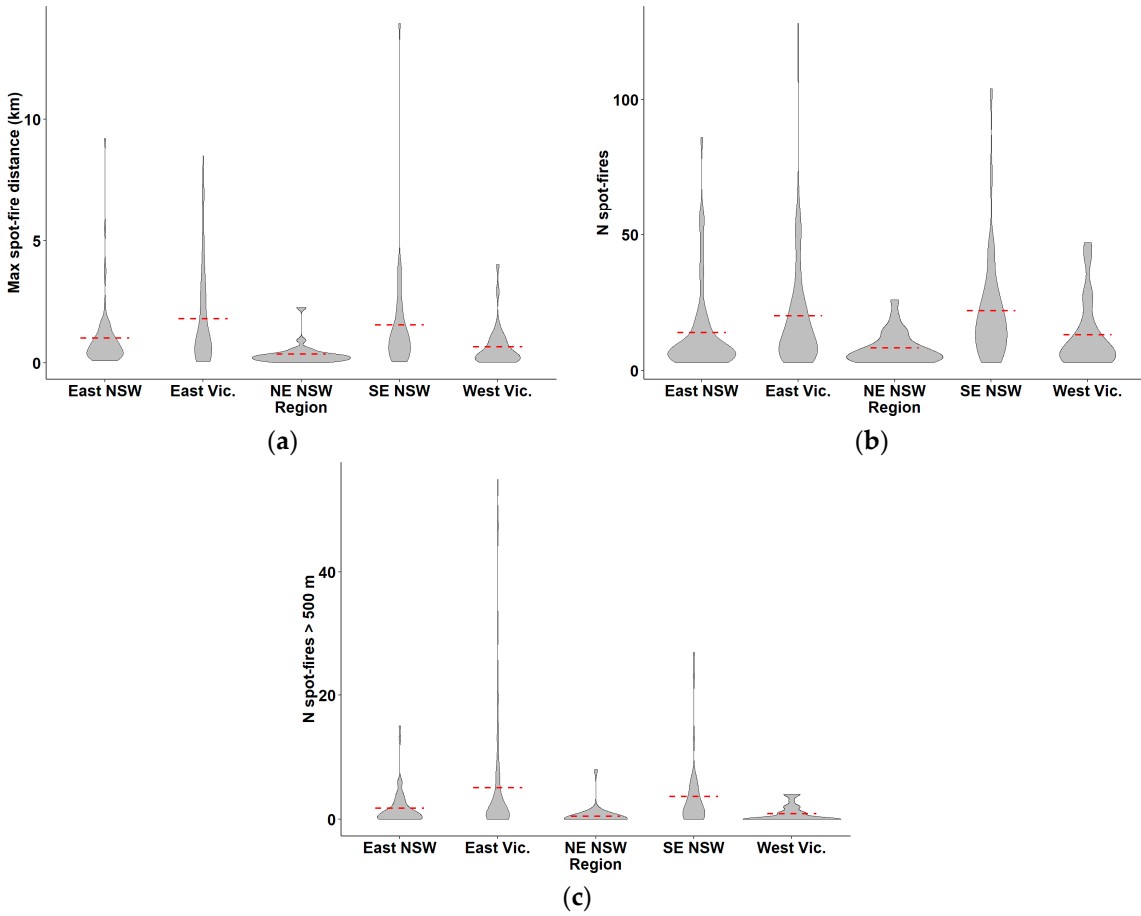

**Figure 10.** Violin plots of region vs. maximum spot fire distance (**a**), number of spot fires (**b**), and number of spot fires > 500 m (**c**). The red dashed line is the mean group value.

### 3.5. Regional Spotting Distributions

Exponential was the most common type of distribution in every region (Table 3). East Victoria had the highest percentage of the multi-modal distribution type (25.5%), followed by SE NSW (18.9%) and east NSW (9.1%). West Victoria and NE NSW had very low proportions of other-distribution and multi-modal (0–6.7%). Chi-Squared indicated a significant relationship between distribution type and region (i.e., the variables were not independent; $p < 0.01$).

**Table 3.** Percentage of source fires of each distribution type by region.

|  | Multi-Modal | Exp. | Other |
|---|---|---|---|
| **East Vic.** | 25.5% | 64.9% | 9.6% |
| **SE NSW** | 18.9% | 81.1% | 0.0% |
| **East NSW** | 9.1% | 81.8% | 9.1% |
| **West Vic.** | 6.7% | 93.3% | 0.0% |
| **NE NSW** | 0.0% | 95.8% | 4.2% |

### 3.6. Regional Environment

East Victoria and SE NSW source fires had the highest mean and largest range of annual rainfall values (Figure 11). West Victoria source fires had the lowest annual rainfall. East Victoria and east NSW source fires had the largest mean canopy height values, while NE NSW and west Victoria had the lowest.

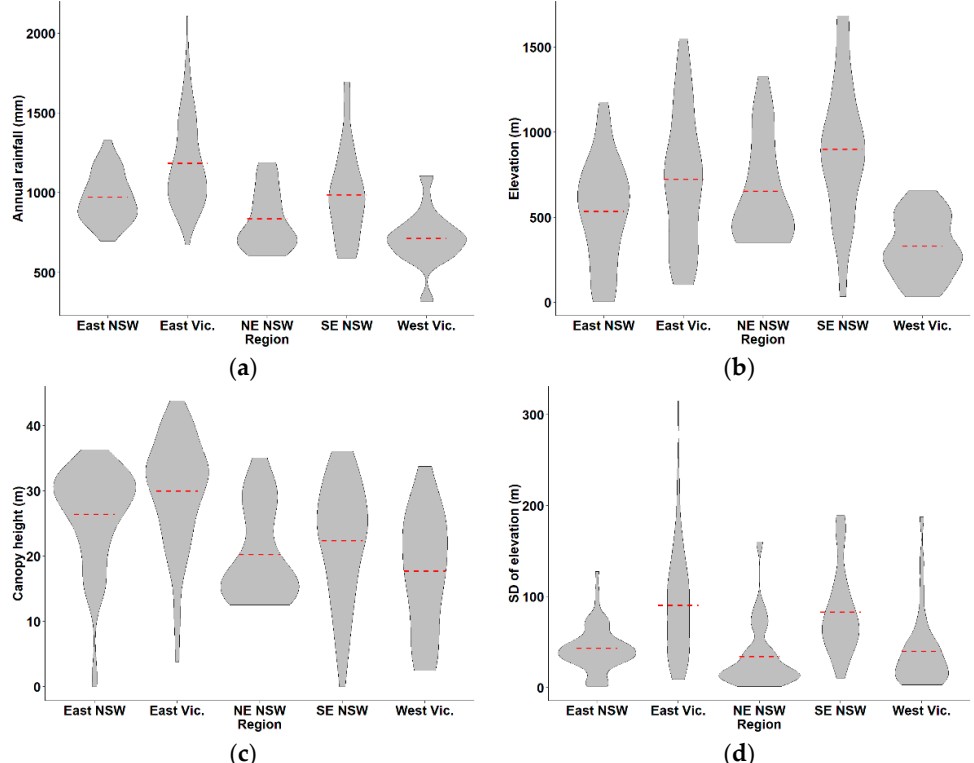

**Figure 11.** Violin plot comparison of regionally summarized mean source fire rainfall (**a**), mean source fire elevation (**b**), mean source fire canopy height (**c**), and source fire standard deviation of elevation (**d**) values. The red dotted line is the mean group value.

East Victoria and SE NSW had the highest mean elevation and were the most rugged (high mean values of SD of elevation). East NSW and NE NSW both had mean elevation values above 500 m, but less variable elevation (Figure 11). West Victoria had both low mean elevation and low standard deviation of elevation, indicating that most source fires there were burning in relatively flat terrain.

East Victoria had the highest proportion of ribbon bark in source fire areas (61.3%), and also 14.2% of the combined ribbon–stringy bark class (Figure 12). West Victoria fires had the highest proportion of stringy bark dominant fires (35.7%), while all other regions had <12% stringybark. While SE NSW had relatively low proportions of stringy bark (12.4%) and ribbon bark (16.3%), there was a large area of combined ribbon–stringy dominance (32.1%). NE NSW (88.9%) and East NSW (72.5%) source fires were mostly of the other bark type dominant category.

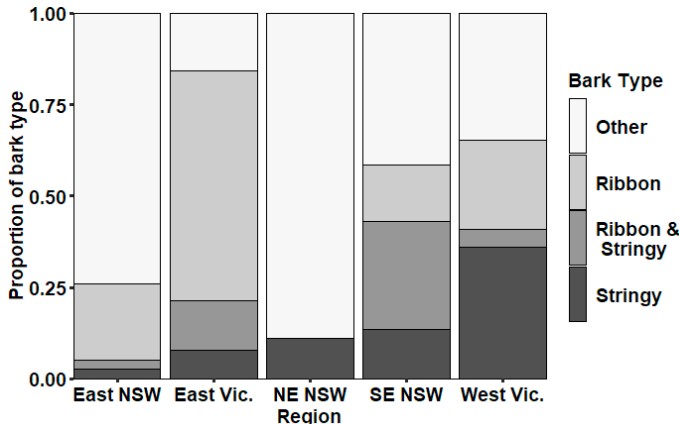

**Figure 12.** Proportion of different bark types by region. Sampled from source fire areas for each region.

## 4. Discussion

### 4.1. Spot Distance and Number

From our dataset of line scans covering the period 2002 to 2018, we have produced a large dataset of spot fire distance distributions, containing 4219 spot fires from 251 source fires. The patterns found show that spotting in wildfires of southeast Australia is mostly over short distances (median = 0.1 km); however, many instances of significant longer-distance spotting were found (90th%le = 957 m, 99th%le = 3.9 km). A total of 191 spot fires from 48 source fires were over 2 km and 18 spot fires from 11 source fires were over 5 km.

While our data indicate a moderate to strong positive correlation between spotting distance and numbers, there were many source fires that spotted long distances but had few spot fires overall (i.e., flatter distributions with isolated long-distance spot fires, e.g., Figure 3, Appendix A). Different sets of environmental conditions may determine whether a source fire produces mass spotting rather than isolated long-distance spotting. For example, mass spotting would require a large amount of firebrand material, whereas isolated long-distance spotting may only require a small amount of aerodynamic firebrand material under conducive weather and fire conditions. The drivers of such differences are important to investigate, as, operationally, wildfires that produce a few very long-distance spot fires need a different response strategy from that of wildfires that produce dense short- to medium-distance spotting.

### 4.2. Distributions

The exponential distribution of spot fire distances was the most commonly produced overall and in each region. However, the multi-modal type (and sometimes the other-distribution type) was associated with longer maximum spotting distances, high N spot fires > 500 m values, and had significantly higher mean values for all three spotting measures, indicating the longest-distance spot fires in our data were usually isolated or in a small clump.

Our results indicate that right-skewed firebrand distributions (e.g., exponential) provide a good approximation of spot fire distributions in most wildfires, but may underestimate occurrence of long-distance spotting. Many published modelled firebrand distributions are developed from small-scale experiments over several metres [23–25] or are mathematical models of firebrands landing over several hundred metres [19–21,45]. Such models have various limitations that can include assuming a wildfire is burning on flat ground, standard-size firebrands (e.g., small cylinders or spheres), and no plume interaction, and do not model ignition probability after firebrands land to calculate the spotting distribution [9]. Such limitations, often necessary due to computational constraints, restrict their application to our fires for which we measured spotting distributions. For example, most of the wildfires with higher levels of spotting in our study (particularly in east Victoria, SE NSW, and east NSW) burnt through complex terrain and through fuels that produce firebrands of a large range of shapes and sizes, including some ribbon bark types up to several metres long [4].

A greater understanding of the drivers of spotting is still required in order to produce more accurate operational models that predict maximum spotting distance and to produce models that realistically predict distribution of spot fires (including our multi-modal or other-distribution over long distances). Many factors can potentially influence spotting distributions. In the event that firebrands happen to be evenly dispersed, a mixture of suitable and unsuitable receiver fuels (e.g., wet valley and dry ridges) could result in patchy ignitions and distribution of spot fires (e.g., multi-modal). However, uneven distribution of firebrands could result after variation at the initial stage of firebrand generation. Variation in firebrand number and type likely arises from variation in fuels (tree coverage, species, mixed fuel/bark types), topography, weather (temporal and spatial), and the source fire behaviour (e.g., size [3]). Further complexity arises during the transport phase, e.g., wind-blown firebrands landing close to the source fire and plume-lofted firebrands landing at longer distances. Recent work suggests that fire–atmosphere interaction is highly influential and may produce firebrand distributions that

more realistically reflect our observations of spotting (e.g., longer-distance multi-modal distribution). For example, Bhutia et al. [39] found that a fire–atmosphere coupled model produced more stochastic firebrand dispersal patterns over longer distances than non-coupled. Thurston et al. [8] modelled firebrand dispersal over many kilometres and found that by incorporating plume turbulence, localised weaknesses (leading to firebrand deposition) and updrafts (leading to continued firebrand lofting) were created, leading to greater spread and maximum distances of firebrand landing positions than non-turbulent plumes. Spot fires that produce more spot fires would also create variation in the overall spot fire distribution. It is likely that these "second-order" spot fires ignite close to their parent spot-fire, which is generally small and thus has lower dispersal potential (e.g., small plume and energy output), particularly in relation to the source fire, which has more potential to produce the longer-distance spot fires (more energy, larger plume, etc.).

More data collection at wildfires is necessary to further understand the complex processes that result in different types of distributions. Better understanding of the amount and type of firebrand material produced under different conditions and further understanding of how different firebrands are transported is necessary (including the role of the plume). Utilising infrared cameras (e.g., helicopter-mounted Forward Looking Infrared (FLIR) camera) to track firebrands from generation to landing and spot fire ignition may be a useful approach.

*4.3. Regional Variation*

We found significant regional variation in spotting for the period 2002 to 2018, with east Victoria being the most prone to spotting. In particular, east Victoria had the highest levels of spotting associated with more extreme wildfires, long-distance spotting, and high N spot fires > 500 m. This result is in agreement with previous anecdotal reports suggesting that the longest-distance spot fires reported for southeast Australia have also occurred in east Victoria: 30–35 km in the Kilmore East Fire of Black Saturday 2009 [10] and 29 km during the 1965 wildfires in east Victoria [46]. SE NSW was slightly less spotting-prone and had the longest single spot fire in this study (13.9 km, Wandoo fire 2006). East NSW also had substantial spotting, except for lower levels of N spot fires > 500 m. Spotting occurred in NE NSW and west Victoria at substantially lower levels than in the other regions.

Several interrelated factors that vary regionally are likely to have influenced the spotting process to enhance spotting behaviour in some regions. The Great Dividing Range runs at least partly through all of our regions, but peaks in SE NSW (Mount Kosciuszko, 2228 m) and east Victoria (Mount Bogong, 1986 m). Our data suggest that elevation and particularly ruggedness may be important, particularly to high N spots > 500 m. The two regions that produced the highest N spots > 500 m had fires burning in the most rugged terrain. NE NSW and East NSW had low levels of N > 500 m and, while being at relatively high mean elevations, fires burning there were in much less rugged locations (lower SD of elevation). The ruggedness of the area could create turbulent local winds that may enhance spotting behaviour [47]. The difference in elevation between where a firebrand is launched (e.g., a ridge top) and where it lands (e.g., valley) may also enhance spotting distances [44].

The Great Dividing Range is associated with higher rainfall, which allows for taller eucalypt forests to grow, although this is also affected by elevation (forest height declines towards alpine tree line) and soils. Our analysis of canopy height and bark types suggest that east Victoria supports particularly tall forests with a high proportion of ribbon bark (e.g., Ribbon Gum, *Eucalyptus viminalis*), although stringy bark types also grow (e.g., Messmate, *Eucalyptus obliqua*). These tall mountain forests can burn very intensely during dry summers, particularly during harsh episodic droughts. The two driest (lowest annual rainfall among source fire locations) regions in our study were the least spotting-prone (NE NSW and west Victoria). Low water availability limits forest growth, producing shorter and sparser forests and woodlands (e.g., Mallee, Heathy Forest) [48]. This appears to be an important factor. West Victoria had a relatively large proportion of ribbon bark types, and also stringy bark (e.g., Brown Stringybark, *Eucalyptus baxteri*); however, it had low levels of spotting, likely because fewer firebrands are produced from the smaller tree forms (including some Mallee species).

Firebrand transport models suggest that wildfire intensity and plume strength influence lofting height of firebrands, while ambient winds influence downwind transport and the degree of plume tilt [7]. The likelihood of intense plume development is possibly linked to fuel biomass (taller vegetation means more energy to release) and size of the source fire [3]. Several examples of deep convective plumes, from some of the largest wildfires recorded in southeast Australia, have occurred in east Victoria (e.g., Black Saturday) and SE NSW (Alps 2003). However, extreme wildfires with deep convective plumes have also occurred in the least spotting-prone regions of our study (e.g., Pilliga 2006 in NE NSW, Grampians 2014 in west Victoria [49]), but very long-distance or mass spotting (> 500 m) has not typically been reported. It is likely that lower firebrand fuel availability limits the amount and distance of spotting in these regions when deep convective plumes do develop.

It should be noted the wildfires included in this study were all prior to the most recent summer (2019–2020) in Australia, in which large wildfires occurred in our regions of NE NSW, East NSW, SE NSW, and east Victoria over several months. Initial inspection of some 2019–2020 line scan data indicate significant spotting fires across all of these regions, including long-distance spotting in NE NSW (some maximum distances > 10 km, a few fires with 5–10 spot fires > 500 m). The wildfires in the NE NSW region during the 2019–2020 summer appear larger and more widespread (incorporating a wider range of forest types) than the wildfires included in our 2002–2018 data. Initial inspection also indicates further significant spotting events in our most spot-prone regions of SE NSW and east Victoria (some fires with > 10 spot fires > 500 m, at least one fire with > 50 spot fires > 500 m). A detailed analysis of the recent data, and new data as they are collected, would be required to update the regional spotting patterns.

Some significant spotting wildfires may have also occurred historically that were not captured in our line scan data, including in our less spot-fire-prone regions. For example, despite having no observations of very long-distance spotting in west Victoria in our study, Rawson et al. [15] reported maximum distances of at least 10 to 12 km and up to 25 km in the "East Trentham/Macedon Fire" (part of the 1983 Ash Wednesday wildfires). This wildfire burnt through a part of western Victoria with taller spotting-prone forest (including Messmate, *E. obliqua*, Candlebark, *E. rubida*) around 30 km northwest of Melbourne. Our regional groupings are too coarse of a scale to capture important local variation of fuel or topography. While we have identified regions more prone to spotting, it is vital to consider the local variations in fuel, topography, and weather patterns to further understand spotting [3].

Continued capture of high-quality wildfire observations is vital to increase our understanding of spotting. Line scanning currently provides the best operational mapping of wildfires, and the continued availability and utilisation of these data for research on spotting are currently irreplaceable. When captured at high temporal frequency, details of wildfire spread and spot fire coalescence can be measured. Such observations could be added to a database to provide a unique resource for empirical research and validation of spotting and wildfire spread models.

## 5. Conclusions

Our analysis found most spotting to be over a short distance (median = 0.1 km), with long-distance spotting occurring less frequently; 10% of spot fires were beyond 1 km and 2% were beyond 3 km. Long-distance spotting appears to be mainly associated with our multi-modal distribution, and long-distance spotting was found to occur in some cases despite the overall number of spot fires being relatively low. This has implications for spot modelling, as use of an exponential or right-skewed distribution for spotting modelling model may not realistically predict isolated long-distance spot fires.

Our study has highlighted important regional variation in spotting in wildfires of southeast Australia. Mass spotting and very long-distance spotting was mostly restricted to wildfires in east Victoria, SE NSW, and, to a lesser extent, in east NSW. The region with the most intense spotting also had the highest annual mean rainfall, was most rugged, and had the tallest canopy height and highest proportion of ribbon bark, suggesting that a combination of environmental drivers are important

for spotting. The regional analysis should be updated as new data become available (e.g., from the 2019–2020 summer).

Access to operationally acquired line scan data is vital to improve our empirical understanding of spotting and wildfire behaviour in general. Access to line scans has allowed us to create a useful empirical dataset of spotting and spotting distributions. While mainly for operational purposes, there is high research value that should encourage continued development and expansion of such data collection methods. Wildfire observations need to be expanded to include plume and firebrand observations to further understand spotting.

**Supplementary Materials:** The following are available online at http://www.mdpi.com/2571-6255/3/2/10/s1, Table S1: source_fire_sample_points; Table S2: source_fire_summary_data; Table S3: spot_fire_distances.

**Author Contributions:** Conceptualization, M.A.S.; methodology, M.A.S. and O.F.P.; formal analysis, M.A.S.; writing—original draft preparation, M.A.S.; writing—review and editing, M.A.S., O.F.P., R.A.B., and J.J.S.; supervision, O.F.P., R.A.B., and J.J.S. All authors have read and agreed to the published version of the manuscript.

**Funding:** The provision of a PhD scholarship to Michael Storey from the Bushfire and Natural Hazards Cooperative Research Centre is gratefully acknowledged. Jason Sharples is also supported by project funding from the Bushfire and Natural Hazards Cooperative Research Centre.

**Acknowledgments:** Data for this research were provided by the NSW Rural Fire Service and Victorian Department of Environment, Land, Water, and Planning. OpenStreetMaps (OSM) basemaps were used for Figures 1 and 9. OSM copyright and licence available online: https://www.openstreetmap.org/copyright.

**Conflicts of Interest:** The authors declare no conflict of interest.

## Appendix A  Map Examples of Source Fire and Spot Fire Polygons of Each Spotting Distribution Type

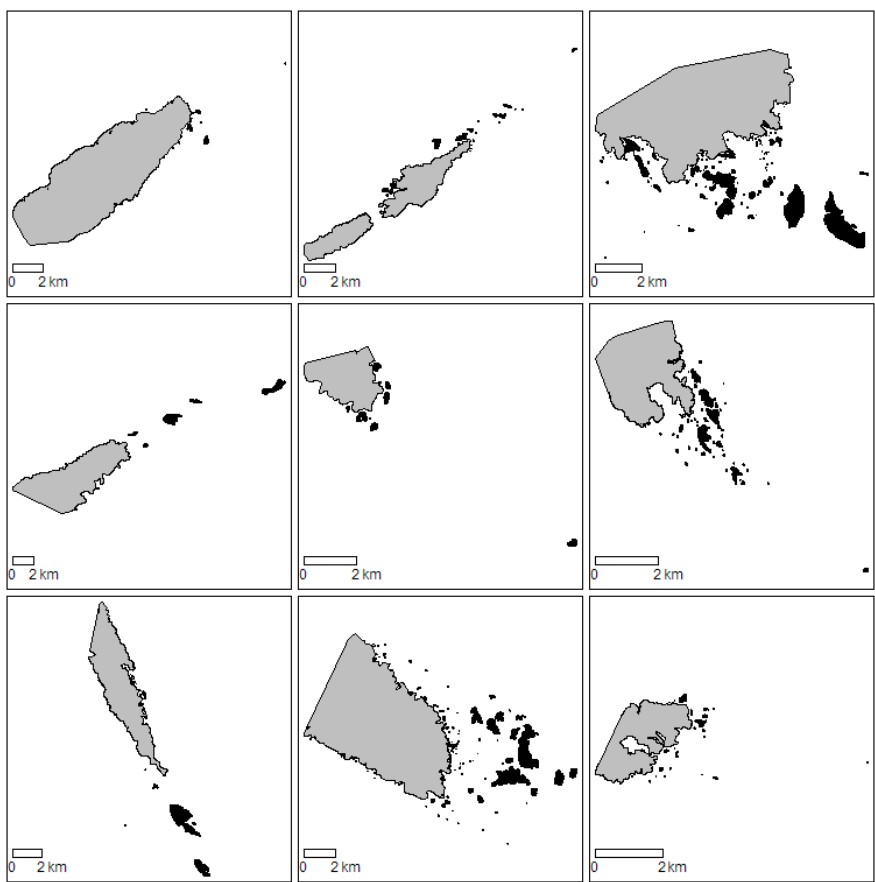

**Figure A1.** The nine multi-modal distribution fires with the longest maximum spotting distances. Source fire is in grey, spot fires are in black.

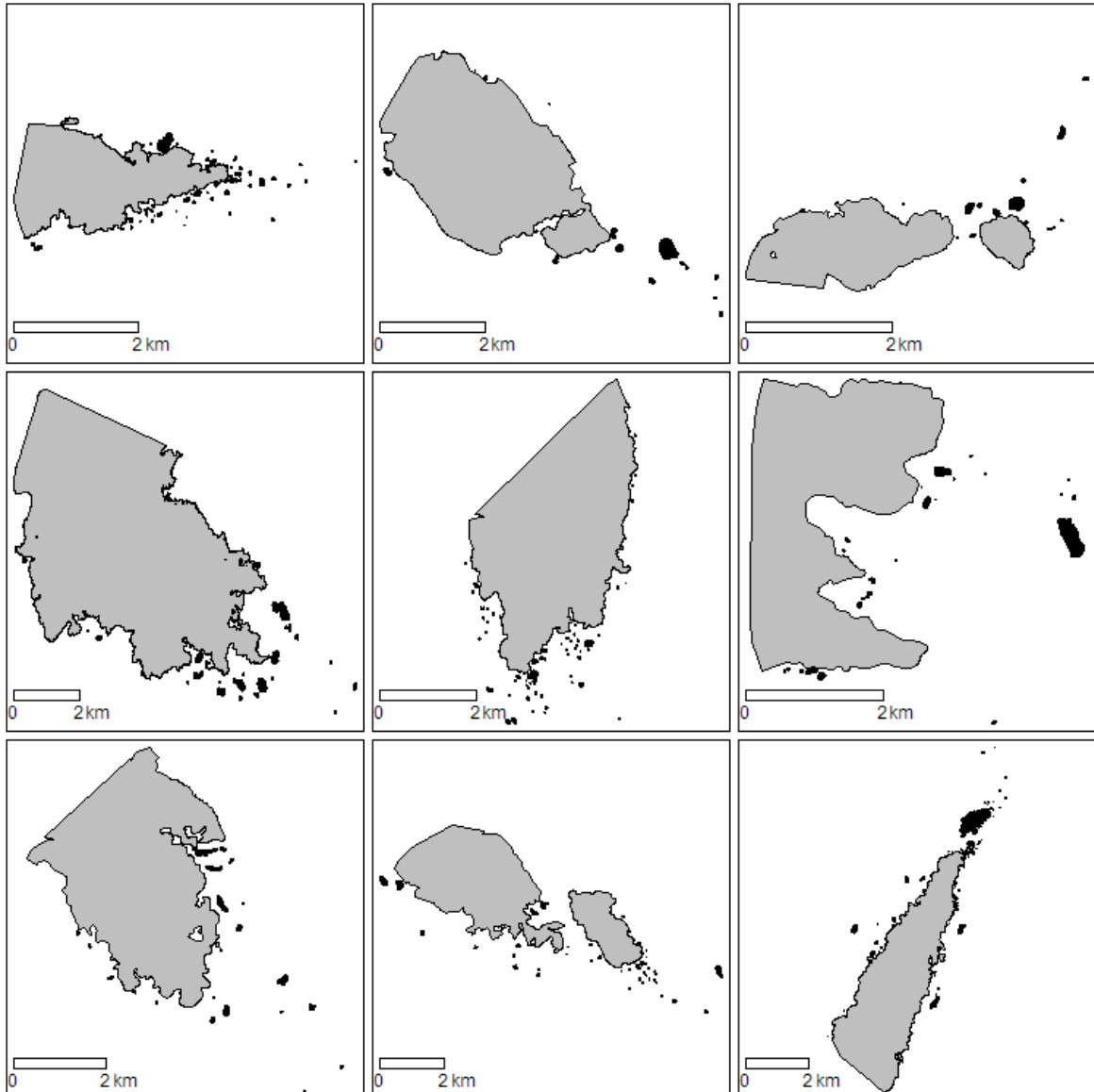

**Figure A2.** The nine exponential distribution fires with the longest maximum spotting distances. Source fire is in grey, spot fires are in black.

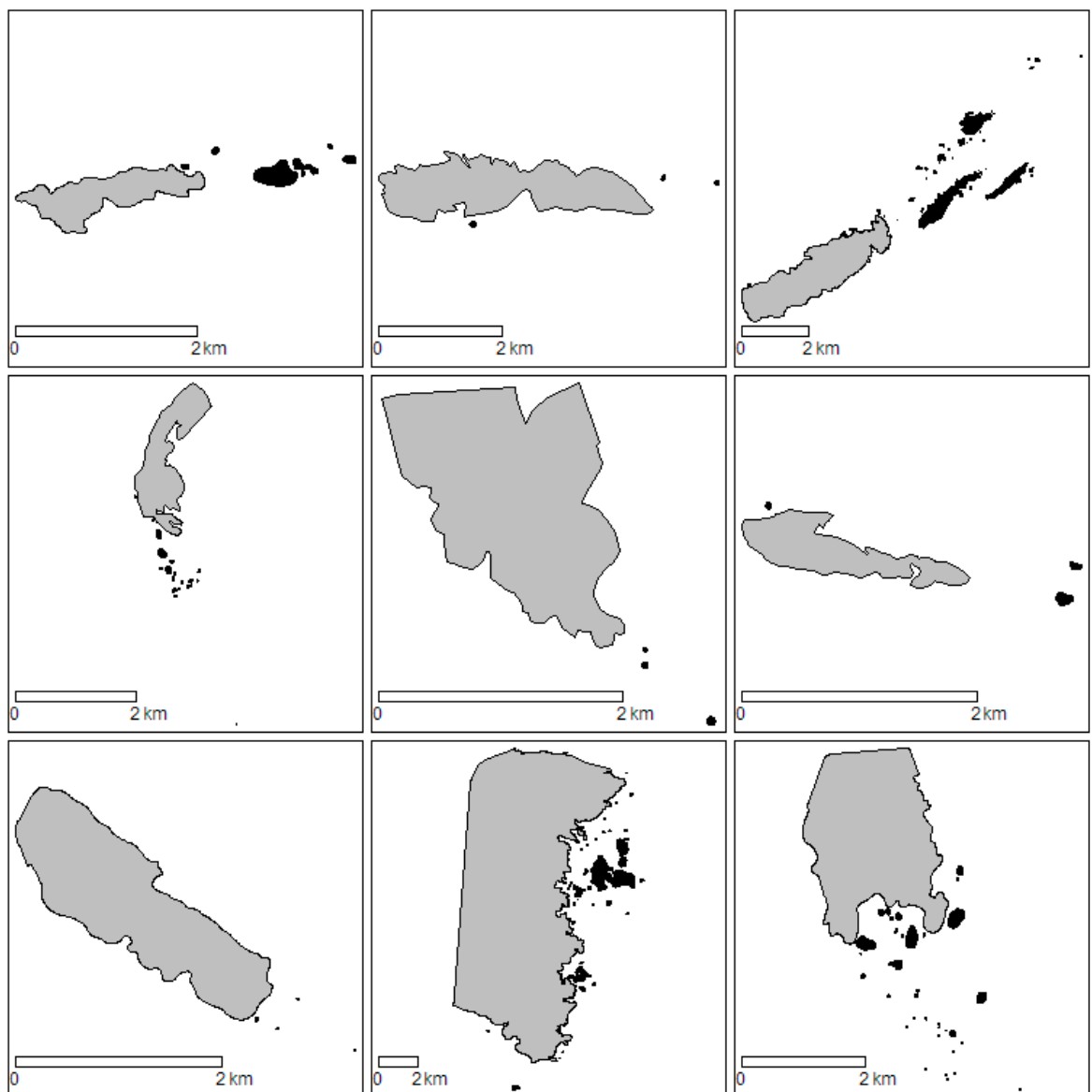

**Figure A3.** The nine other-distribution fires with the longest maximum spotting distances. Source fire is in grey, spot fires are in black.

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
