# Peer review of "Analysis of Variation in Distance, Number, and Distribution of Spotting in Southeast Australian Wildfires"

_fire, doi:10.3390/fire3020010_

Round 1
Reviewer 1 Report
I think this is a continuation of excellent work on spot fire analysis by this team. I have only a few minor suggestions:
- A major factor for spot fire distance is governed by wind (as mentioned in the paper), but this doesn't seem to come into the analysis at all? I think at least something should be said about the fire conditions for the sample fires used. I am guessing as these were line scanned they were all major fires with a corresponding high danger weather conditions.
- Figure 5 would look much better on a semi-log plot. A governing power law could also easily be derived from this data.
- Is long distance spotting mainly associated with multi-modal fires because these fires have generally started large fires ahead of the front (as shown in the images)? If so, this would just be an exponential distribution captured at a later time? I think this needs to be covered in the discussion.
- L216/L218 I think the figure numbers need checking - should be 14 and 13?
Author Response
Thanks to the reviewer for their review and suggestions. Our responses to each are below. Author responses in italic text.
Comments and Suggestions for Authors
I think this is a continuation of excellent work on spot fire analysis by this team. I have only a few minor suggestions:
- A major factor for spot fire distance is governed by wind (as mentioned in the paper), but this doesn't seem to come into the analysis at all? I think at least something should be said about the fire conditions for the sample fires used. I am guessing as these were line scanned they were all major fires with a corresponding high danger weather conditions.
I have added some summary stats for weather values among the fires used in this study on L179/180:
“The 251 source fires included spanned a wide range of fire weather and behaviour (for all fires wind gusts (km/h) ranged 14 to 81 (mean 37), temperature ranged (Celsius) 12 to 42 (mean 28), relative humidity (%) ranged 7 to 63 (mean 27) [2]).”
- Figure 5 would look much better on a semi-log plot. A governing power law could also easily be derived from this data.
The main aim with this plot is to just show that most spot-fires overall were short distance and I’d prefer not to transform the data here. However, having another look I do think this plot displays the data better as a histogram (as it shows gaps in spot-fire distance observations), so I have replaced the original with a histogram here.
- Is long distance spotting mainly associated with multi-modal fires because these fires have generally started large fires ahead of the front (as shown in the images)? If so, this would just be an exponential distribution captured at a later time? I think this needs to be covered in the discussion.
Long-distance spotting is probably mainly associated with multi-modal through some combination plume/wind and fuel at the source fire and fuel at receiver sites. We discuss these potential influences L168 -182, but we couldn’t be much more detailed than this without better data.
It is likely that spot-fires closer to the source-fire are causing more spot-fires at the longer distances, although from the scans it appears that if spot-fires are igniting more spot-fires that they are close proximity, and the very long distance spots are coming from the source-fire (although there is no sure way to tell just from the scans).
In addition to mentioning this in the methods L157-158, I added a sentence to the discussion about the spot-spot ignition (L383-387):
“Spot fires that produce more spot fires would also create variation in the overall spot fire distribution. It is likely these “second-order” spot fires ignite close to their parent spot-fire, which is generally small, thus has lower dispersal potential (e.g. small plume and energy output), particularly in relation to the source fire that has more potential to produce the longer distance spot fires (more energy, larger plume etc.).”
- L216/L218 I think the figure numbers need checking - should be 14 and 13?
I have rearranged the wording here as it was a bit confusing before. The meaning was to refer the reader to both the Appendix (which has many examples) and a figure in the main text of that distribution type. It now reads:
“(e.g. Appendix A, also the fires in Figure 2).” and “(e.g. Appendix A, also the fire in Figure 3).”
Reviewer 2 Report
The aims of the paper are [1] to present data about fire spotting (numbers, distance distributions, maximum distances), which was remotely measured on hundreds of fires in Australia, [2] to characterize patterns in that data with descriptive statistics and [3] to further analyze that data using various statistical tests to identify significant differences in terms of variables including region, physiography and fuel characteristics.
The main contributions of the research are [1] the collection and characterization of patterns in fire spotting data from many real fires, which is otherwise seriously lacking in the literature, [2] giving a literature review of spotting data and models and [3] providing a comprehensive discussion of the shortcomings of this data and analysis as direction for further data collection and analysis.
General comments:
It has been a rare pleasure to read and review this paper, because [1] the research was so well-designed, [2] the paper is so well organized and [3] the paper is so well-written. As I read the paper, I collected notes for various shortcomings that I thought of, but the authors had identified those and others in the comprehensive discussion.
This paper will be very valuable to researchers who wish to model the behaviour of fire spotting and to anyone who wishes to measure fire spotting in the field and analyze the data.
Specific comments:
The paper is well-written and clear enough as is. I believe the copy editor will improve any language, if necessary, and catch items such as Figure 10 being mis-numbered. There are a few descriptive passages where the writing is complex, but I worked through it all and found it to be accurate.
Author Response
Thank you for your review and encouraging comments. I will work with the copy editor to make sure figure numbers etc. are correct.